# Real-Time Analysis of Oxygen Gradient in Oocyte Respiration Using a High-Density Microelectrode Array

**DOI:** 10.3390/bios11080256

**Published:** 2021-07-29

**Authors:** William Tedjo, Yusra Obeidat, Giovana Catandi, Elaine Carnevale, Thomas Chen

**Affiliations:** 1Department of Electrical and Computer Engineering, Colorado State University, Fort Collins, CO 80523, USA; contact@wtedjo.com; 2Electronic Engineering Department, Yarmouk University, Irbid 21163, Jordan; yusraobeidat@yahoo.com; 3Department of Biomedical Sciences, Colorado State University, Fort Collins, CO 80523, USA; giovana.catandi@gmail.com (G.C.); elaine.carnevale@colostate.edu (E.C.); 4School of Biomedical Engineering, Colorado State University, Fort Collins, CO 80523, USA

**Keywords:** CMOS biosensor, microelectrode array, electrochemistry, microfluidics, oxygen concentration gradient, oxygen flux, oxygen consumption rate

## Abstract

Physiological events related to oxygen concentration gradients provide valuable information to determine the state of metabolizing biological cells. The existing oxygen sensing methods (i.e., optical photoluminescence, magnetic resonance, and scanning electrochemical) are well-established and optimized for existing in vitro analyses. However, such methods also present various limitations in resolution, real-time sensing performance, complexity, and costs. An electrochemical imaging system with an integrated microelectrode array (MEA) would offer attractive means of measuring oxygen consumption rate (OCR) based on the cell’s two-dimensional (2D) oxygen concentration gradient. This paper presents an application of an electrochemical sensor platform with a custom-designed complementary-metal-oxide-semiconductor (CMOS)-based microchip and its Pt-coated surface MEA. The high-density MEA provides 16,064 individual electrochemical pixels that cover a 3.6 mm × 3.6 mm area. Utilizing the three-electrode configuration, the system is capable of imaging low oxygen concentration (18.3 µM, 0.58 mg/L, or 13.8 mmHg) at 27.5 µm spatial resolution and up to 4 Hz temporal resolution. In vitro oxygen imaging experiments were performed to analyze bovine cumulus-oocytes-complexes cells OCR and oxygen flux density. The integration of a microfluidic system allows proper bio-sample handling and delivery to the MEA surface for imaging. Finally, the imaging results are processed and presented as 2D heatmaps, representing the dissolved oxygen concentration in the immediate proximity of the MEA. This paper provides the results of real-time 2D imaging of OCR of live cells/tissues to gain spatial and temporal dynamics of target cell metabolism.

## 1. Introduction

Oxygen consumption measurement at the cellular level has been of interest in physiological studies. Cell oxygen consumption affects cell function and its viability [1,2]. In the cell metabolic pathways, the mitochondrial oxidative phosphorylation (OXPHOS) process yields a chemical complex adenosine triphosphate (ATP), which is a high-energy molecule required to maintain cellular activities [3,4]. Therefore, the ability to image oxygen consumption and gradient in real-time is highly desirable to better understand the role of oxygen, such as in the metabolic progress of non-communicable diseases (e.g., cancer, neurodegenerative, autoimmune, and cardiovascular diseases) [5], mechanism of tumor cell growth [6,7], embryo morphogenesis [8], and development of assisted reproductive technology (ART) [9,10,11,12]. The oxygen consumption rate (OCR) provides insight into a cell’s development stage and prognosis and is potentially useful for rapid therapy treatments [13,14]. Meanwhile, in the development of an oocyte or early embryo, monitoring partial pressure of oxygen (pO_2_) in the cell’s medium over time provides the OCR information of the cell during mitochondrial OXPHOS, which is one of the predictors of a cell’s viability during its maturation and its effect to various therapies in assisted reproductive technology (ART) [12,15,16]. A practical pO_2_ measurement method with sufficient sensitivity and spatial and temporal resolutions will greatly advance treatments and drug development in the medical field.

Developments of oxygen sensing methods for a microenvironment are generally tailored to specific applications. Desirable features of a pO_2_ imaging system include high spatial and temporal resolutions, low limit of detection, non-invasiveness to bio-samples, sensor reusability, repeatability, and compatibility to other types of sensing (e.g., temperature, pH, chemical, impedance). The existing methods of oxygen sensing include electrochemical methods (amperometric and potentiometric), optical methods (fluorescence and phosphorescence), and magnetic and paramagnetic resonance imaging methods (MRI and EPRI). Both MRI and EPRI methods offer three-dimensional (3D) non-invasive in vivo sensing, while requiring contrast agent and complex magnetic resonance instrumentations to perform [17,18,19,20]. The optical method has been well-established and commercialized over the years for in vivo and in vitro tissue and cell culture imaging [21,22,23]. The latest development of optical pO_2_ imaging enables scientists to perform micrometer spatial resolution intracellular imaging [24], offers in vivo imaging [25], and allows 3D cell cultures imaging using oxygen sensing microbeads [26]. Despite its advancements, the optical method has drawbacks of its own (e.g., phototoxicity, photobleaching, and photostability) when used for real-time in vitro applications [27]. In contrast to other methods, the electrochemical method offers a distinct advantage to measure the concentration level of target chemical species without the need for bio-sample pretreatment or an optical probe. Moreover, the electrochemical method has been proven to allow simultaneous detection of multiple chemical contents, such as neurotransmitters [28,29] and metabolic markers [30,31,32].

The electrochemical method for pO_2_ sensing was initially performed, and its capability in one-dimensional (1D) data points of biological samples has also been demonstrated [12,33,34,35,36,37,38,39,40]. Although measurements of oxygen consumption rate were done successfully, the 1D electrode system can only measure the oxygen concentration in the vicinity of the cell without spatial information [33,41]. Oxygen concentration gradients with sufficient spatial information illustrate the flux of oxygen through the medium to the underlying cell. Therefore, the two-dimensional (2D) gradient profiles could give a better understanding of the cell’s metabolic activity with respect to the rate of oxygen delivery to the cells [42], especially when the cell distribution is heterogeneous and/or when the gradients are uniform in different directions. The 2D microelectrodes array (MEA) integrated within an electrochemical imaging system can provide an attractive means of measuring OCR based on the cell’s 2D oxygen concentration gradient. Capturing the pO_2_ gradients has been attempted in the past with a small number of microelectrodes in a circular configuration [43]. An earlier attempt at pO_2_ imaging using a complementary-metal-oxide-semiconductor (CMOS)-based MEA sensor was also presented in 2015 [44]. However, the device was only characterized in non-biological flow injection experiments at 250 µm spatial resolution and 400 measurement pixels.

This paper presents the results of pO_2_ imaging of in vitro experiments at high spatial and temporal resolutions using the electrochemical method. The electrochemical sensor system is a continuous effort from our previous works in chemical sensing employing microelectrode in three-electrode configuration with Working Electrode (WE), Reference Electrode (RE), and Auxiliary Electrode (AE) [45,46,47,48]. The results from our microelectrode geometry optimization for maximizing sensor output signals were used in this design [45]. Initial investigations were completed to confirm the viability of performing electrochemistry in dense MEA [46,47]. The system presented in this paper was recently developed and evaluated to offer higher spatial and temporal resolution with improved sensitivity to target chemical analytes. The system employs a custom-designed CMOS microchip with 16,064 surface MEA at 27.5 µm spatial resolution and with a frame rate of up to 4 Hz temporal resolution [48]. In our work employing microelectrode structures, a protective membrane commonly employed in Clark-type electrodes [37,49] was not used to allow integration with the standard CMOS manufacturing technology. To illustrate its capability in capturing pO_2_ gradients, sets of in vitro experiments on bovine cumulus-oocytes-complexes (COCs) were performed to generate 2D pO_2_ images to capture COCs’ pO_2_ consumption in real-time. To calculate flux density and consumption rate in the high-density MEA, our results were analyzed according to a previously established work of pO_2_ gradient simulations and measurements [43]. A custom microfluidic support system was also designed and fabricated to allow reliable and repeatable sensing of in vitro experiments. In combination with a robust microfluidic sample handling, this work offers unique insights into the metabolic response of small biological samples with high spatial and temporal resolutions.

## 2. Materials and Methods

### 2.1. Sensor System Overview

As described and illustrated in Figure 1A, the electrochemical sensor system comprises a swappable CMOS microchip, three stacks of supporting printed circuit boards (PCBs) (i.e., Sensor PCB, the Shield PCB, and Control PCB), a microfluidic support system, and a pressure plate.

The electrochemical sensor system was divided into three PCBs according to their specific functionalities. The Sensor board holds the 280-pin socket for swapping CMOS microchips in and out of the system, precision low-noise power supply regulators for the entire system, and a programmable potentiostat. The Control board receives, processes, and transmits digital signals through the USB protocol to a host PC. The Shield board provides regulated thermal dissipation and electromagnetic isolation between the sensitive analog components on the Sensor board and the noisier digital components on the Control board. Analog outputs from the CMOS microchip are directed to the host PC with a commercial data acquisition board. The control system, environment monitoring, and signal processing occur in the host PC, and the real-time experiment information is provided via a custom user interface. The specification of the electrochemical sensor system is summarized in Appendix A, and the depiction of the user interface on the host PC is presented in Appendix A.

The electrochemical sensor system employs a custom microfluidic system for bio-sample handling and delivery to the surface MEA. A microfluidic system offers a convenient method to safely and reliably handle oocytes/embryos while allowing integrations of sensing electrodes [50,51]. As depicted in Figure 1A and Figure 2, the microfluidic system consists of a thin polydimethylsiloxane (PDMS) (Sylgard™ 184, Dow Corning, Midland, MI, USA) membrane as a channel, a laser-cut PMMA cover, an Indium-tin-oxide (ITO) coated glass, and metal needles as inlet/outlet. The choice of using PDMS as the membrane material is based on its ability to easily form a soft-sealing interface between the hard silicon microchip surface and the polymethylmethacrylate (PMMA) cover. The thin PDMS membrane (0.3 mm) was fabricated to minimize gas diffusion from the external environment into the microfluidic chamber during a typical experiment period in the order of a few minutes. A laser-cut PMMA cover houses an Indium-tin-oxide (ITO) coated glass that functions as AE. This microfluidic system is held down to the CMOS microchip passivation surface using a pressure plate. The pressure plate not only secures and seals the microfluidic channel but also provides an upright opening for optical microscopy access.

Accurate temperature control during sensing is crucial to maintain the normal physiological functions of the bio-sample. The system continuously maintains a target temperature of the MEA surface within 0.5 °C of the desirable physiological temperature (37.0 °C). The heat is generated by the CMOS microchip power dissipation. Hence, the temperature at the surface MEA can be precisely controlled by a feedback system to adjust the power supply voltage of the CMOS microchip.

### 2.2. CMOS Microchip with Microelectrode Array

The CMOS microchip was custom-designed and fabricated in a standard 0.6 µm CMOS process with additional process steps to form the MEA on the microchip surface. This exposed metal layer is coated with layers of Ti, Au, and Pt, with Pt as the outermost layer. The choice of using the 0.6 µm CMOS process is for its availability and ability to allow additional surface MEA post-processing steps. The detailed process of the surface electrode fabrication was discussed in previous publications [45,47].

As shown in Figure 1B, the MEA covers a 3.6 mm × 3.6 mm section at the center of the microchip with the read channel array tightly packed at the top and bottom part of the 19 mm × 19 mm silicon die. There are 80 pairs of Utility Electrode (UE) distributed evenly within the surface area of the MEA. UE is measured at four times larger than WE. It is depicted as small dots forming a circular pattern at the center of the MEA in Figure 1C and is depicted in an 8 × 8 section of WEs array in Figure 1D. Each UE is directly accessible from the CMOS microchip external pins for impedance measurement or electrical stimulation, however, the use of the UEs is beyond the scope of this paper. The UEs are electrically isolated for any experiment result presented in this paper. As illustrated in Figure 1D, a 2.5 µm wide RE encloses individual WEs. Each WEs are in the form of a pair of electrically shorted C-shaped electrodes covering a 17.5 µm × 17.5 µm area, with 27.5 µm distance to surrounding WEs. The process of forming the electrode surface structure for the MEA results in the electrode pattern with 1.5 µm-tall features raised above the passivation layer of the silicon die as shown in Figure 1E.

### 2.3. Microfluidics System and Fabrication

The fabrication process for the microfluidic support system requires only low-cost, off-the-shelf components and a laser cutter. The unassembled parts for the microfluidic cover are depicted in Figure 2A. A 3 mm-thick PMMA sheet is cut to form an 18 mm × 18 mm frame with hole openings for inlet/outlet needles and a square placeholder for the ITO-coated glass. The inlet/outlet needles were extracted from 0.5” 18-ga blunt dispense needles (Loctite^®^ 98399, Henkel, Westlake, OH, USA). The 120 nm ITO-coated glass (ITO-101007-15C, MTI Corp., Richmond, CA, USA) dimension is 10 mm × 10 mm × 0.7 mm, which acts as an AE. The optical transparency of the ITO glass also allows optical imaging of the cells in the enclosed chamber during the experiment. The ITO-coated surface is placed facing down in the middle of the PMMA frame, and a 2 mm-thick PMMA spacer is placed immediately above it to fill the excess space. The microfluidic cover assembly is held together using medical-grade epoxy (EPO-TEK^®^ 301, Epoxy Technology Inc., Billerica, MA, USA). Finally, Ag conductive paint (SPI^®^ Supplies 05001-AB, Structure Probe, Inc., West Chester, PA, USA) is applied to form electrical contact from the ITO-coated surface on the bottom side to the top side of the PMMA cover, where the pressure plate makes a direct electrical contact for AE connection. Using a custom-made PMMA mold, a 0.3 mm-thick PDMS membrane is fabricated and laser-cut to desirable channel patterns matching the microfluidic cover. Figure 2B shows variants of PMMA covers and the PDMS membranes with two and three inlet/outlet, providing options for delivering up to two different fluids for in vitro experiments. The entire microfluidic inlet/outlet needles and channel hold approximately 50 µL, while the 0.3 mm channel height forms a chamber that holds 3.9 µL fluid capacity above the 3.6 mm × 3.6 mm MEA cross-section.

### 2.4. COC Preparation and Handling

Bovine ovaries were obtained within 30 min of post-mortem from a local slaughterhouse. After collection, ovaries were rinsed with room temperature saline solution and transported in the same solution in an insulated container to the Colorado State University Equine Reproduction Laboratory (ERL) within one hour. At the ERL laboratory, ovaries were rinsed three times with sterile saline and held in the same solution at room temperature. COCs were aspirated from follicles ranging between 3 and 8 mm in diameter with a 6 mL syringe and an 18-ga sterile needle, identified with a stereomicroscope at 7× to 10× magnification, picked up using a micropipette, and moved to media, a mixture of G-MOPS™ (Vitrolife, Englewood, CO, USA) handling media and 0.4% bovine serum albumin (BSA) (Sigma-Aldrich, St Louis, MO, USA). COCs were moved to an insulated 1.5 mL microtube filled with media and kept in an incubator (38.5 °C) until the pO_2_ imaging experiment. COC was selected for the assays based on the homogeneity of cytoplasm and compact cumulus cells, as previously described [52]. All pO_2_ imaging experiments presented in this paper were performed within six hours from the COC extraction process from the ovaries. 

### 2.5. Experiment Setups

Oocytes in vitro during experiments require a regulated temperature of 38.5 °C [53]. Using an accurate external reference thermometer (Traceable^®^ 4243, Traceable Products, Webster, TX, USA), the system was calibrated to ambient room temperature to provide 38.0 °C ± 0.5 °C at the MEA surface. To set the MEA surface temperature to a stable point, the CMOS microchip is powered-up for a pre-heat phase for around 30 min before the first experimental run and periodically confirmed with a thermal imaging camera (FLIR^®^ ONE Pro, Teledyne FLIR^®^, Wilsonville, OR, USA). Figure 3A shows a typical arrangement of the pO_2_ imaging experiment. Figure 3B shows a setup with two inlets and one outlet microfluidic support system, and each inlet is being driven by a syringe pump. With the two-inlet configuration, the user can configure different chemical solution delivery at various flow rates within one experiment. The microfluidic setup allows controlled fluid injection using a syringe pump (NE-1000, New Era Pump System Inc., Farmingdale, NY, USA) for delivering and retrieving bio-samples to and from the MEA location on the CMOS microchip.

During each pO_2_ imaging experiment, COCs were moved to 4-well dishes (Nunc™ 179830, Thermo Scientific, Waltham, MA, USA) filled with the media. The storing dish is covered with the accompanying dish lid to allow gas exchange, placed in a temperature-controlled warming plate, and left for approximately 15 min to allow the media pO_2_ to reach the atmospheric saturation point. A 1 mL syringe with 1/32” inside diameter tubing (Tygon^®^ ND 100-65, Saint-Gobain, Malvern, PA, USA) was used to extract the COCs from the storing dish to be injected into the initially dry microfluidic inlet. To retrieve COCs, withdrawal of the syringe was performed using the syringe pump or manually. Figure 3C shows a top view close-up of the transparent ITO AE and a stereomicroscope opening on the pressure plate to optically capture the entire MEA surface. Figure 3D depicts a typical image from a stereomicroscope of COC positioned over the MEA area, ready for pO_2_ imaging. All experiments reported in this paper were conducted with a media mixture of G-MOPS™ and 0.4% BSA.

## 3. Results and Discussion

### 3.1. Electrochemical Oxygen Response

To create low pO_2_ conditions for calibration, a bulk sample of media was degassed in a vacuum chamber and sparged with nitrogen to reach 9.3 µM pO_2_, approximately 1.18% of the maximum 20.95% pO_2_ in ambient air. The oxygen-depleted media was then gassed with oxygen-rich air to reach higher pO_2_. Once the media reached desirable pO_2_, a small volume sample was promptly and gently stored in a syringe to conserve the pO_2_ level. A commercial oxygen meter (Oakton^®^ DO6+, Cole-Parmer, Vernon Hills, IL, USA) was used to confirm seven media pO_2_ samples ranging from 9.3 µM to 165.6 µM, corresponding to approximately 1.18% to 20.95% of ambient air pO_2_. To reduce pO_2_ error, each pO_2_ measurement was done within a minute after each specific pO_2_ sample was created. The flow injection using two-inlet and one-outlet microfluidic channels was employed to switch between two different pO_2_ samples, saturated and reduced pO_2_. The experiments were performed at 37.0 °C ± 0.5 °C, 5003 feet (1525 m) altitude, and 632 mmHg atmospheric pressure.

In a three-electrode configuration using microelectrodes, the amount of electron transfer at the WE due to redox is influenced by several factors such as RE potential, electrode materials, electrode geometry and the relative position of the three electrodes, and the number of target species at the WE surface. Assuming that the design, the materials used for the electrodes, and the electrode voltage setup are known and fixed, the amount of electron transfer at the WE (i.e., the electrical current observed at the WE) due to redox is largely proportional to the concentration of the target species at the WE surface [54]. Assuming the direct four-electron reduction, four electrons are consumed to reduce each oxygen molecule into a water molecule [55]. Therefore, most of the electrical current generated by the reduction process can be represented as oxygen reduction. The reduction potential for oxygen varies depending on factors such as electrode materials used, electrode geometric configuration, the composition of buffer media, temperature, and pH. With strict monitoring of temperature, the use of strictly controlled media buffer in the microfluidic setup, and the fixed electrode geometric configuration, the material used for the electrodes, especially the RE, plays an important role. Inert noble metal, such as Pt, is the preferred material for electrochemical applications. However, a more stable RE material, such as Ag/AgCl, is a better suited redox material for RE. Due to the limitations of the low-cost traditional silicon manufacturing, the same material (Pt) was used for both the RE and WE, resulting in a potential shift in the reduction potential for oxygen. A set of experiments was carried out to determine the reduction potential in our setup. The reduction potential in a range of −0.5 to −0.7 V with respect to the RE potential was obtained, similar to a previously reported work [56]. Others have also observed a varying range of oxygen reduction potential based on the specifics of their experiment and sensor setup [43,57,58]. All amperometry pO_2_ imaging results presented in this paper used −0.6 V reduction potential.

Cyclic voltammetry (CV) experiments confirmed oxygen reduction as depicted in Figure 4A. The results represent individual CV runs at 64 different locations in the MEA. The CV scan rate of 10 V/s was experimentally selected based on consideration of the reduced sensor output current due to WE microscopic size, the sensor electronic system’s signal-to-noise ratio (SNR) requirement, and the maximum current limits to define the lower and upper scan rate boundary, respectively. In comparison, other devices reported in the literature employing microelectrodes have also used CV scan rates range of tens of mV/s to hundreds of V/s based on their system level requirements [29,58,59,60,61,62]. The presented CV results were generated from flow injection experiments with media samples with 9.3 µM pO_2_ (red) and 165.6 µM pO_2_ (blue). As shown in Figure 4A, the large background current (i.e., capacitive current) can primarily be attributed to the charging of the double layer and is proportional to the scan rate. Other electrochemically active ingredients present in the media (a mixture of G-MOPS™ and BSA) can also be oxidized and reduced, which adds to the background current. With a relatively high background current, the relatively small reduction current due to the presence of oxygen can be obtained using background subtraction which has been used widely in the past as the background-subtracted cyclic voltammogram (BSCV) method [63,64]. The background-subtracted signal (shown in green in Figure 4A) indicates that, in this case, the change in current is attributable to oxygen reduction that occurred around −0.55 V. Other similar CV results with significant background current using planar Pt microelectrode have also been observed by others [56,57].

Figure 4B illustrates the sensitivity curve showing responses of media solutions with increasing pO_2_ concentration in linear scale. The curve shows a good linear response (R = 0.98) across all 16-thousands electrodes with 121.8 µM/nA sensitivity. The limit of detection (LoD) can be approximated using the 3σ value of the 9.3 µM pO_2_ readings, which is 165.0 pA, 20.09 µM, 0.643 mg/L, or 15.27 mmHg. The results from the controlled experiment include errors, such as systematic error of MEA pixel-to-pixel variations (e.g., CMOS fabrication imperfection), experimental error of oxygen leakage from the plastic syringe, PDMS, and microfluidic imperfections in delivering specific pO_2_ media. The combined error accounts for up to 0.5% measured error of the 20.95% full scale. 

### 3.2. COC Basal Respiration and MEA Oxygen Reduction

Three experiments were carried out using different potentiostat duty cycles to better understand oxygen reduction dynamics of the MEA enclosed in the microfluidic chamber. The potentiostat duty cycle refers to the duration ratio when the amperometric three-electrode system is at an active state (i.e., V_WE_ vs. V_RE_ = −0.6 V) versus when it is at an inactive state (i.e., V_WE_ = V_RE_ = 0 V).

As plotted in Figure 5A,B, each case of a given potentiostat duty cycle consists of seven sampling segments indicated by the shaded area when the potentiostat is active. Each segment is a group of 10 data points, and each point is the average readings of all 16 thousand WEs signals at one point in time. The saturation point, where the oxygen was mostly reduced, occurred around 75 s, 135 s, and 275 s, for cases #1 to #3, respectively. The fastest rate of oxygen reduction due to active WEs was case #1 (72.5% duty cycle), while the slowest oxygen reduction was case #3 (12.5% duty cycle). These reduction rate differences indicate that oxygen reduction at the WEs only occurred during the time period when the potentiostat was in the active state. However, the amount of reduced oxygen is similar for cases #1 to #3. In addition to cell respiration and diffusion limit during experiments, one unique aspect that determines the trend of dissolved oxygen concentration within the small chamber is the reduction of dissolved oxygen in bulk media, away from the electrode surface. This effect further exacerbates the overall reduction in measured oxygen concentration by the electrodes compared to cases where the transport mechanism is mainly determined by the diffusion limits described by the Cottrell equation. Exponential models were fitted to each case to show this compounding effect in our experiment. The amount of charge measured on each WE can be approximated to be 8.37 nC, 7.27 nC, and 7.02 nC for cases #1, #2, and #3, respectively. Appendix A includes the MEA oxygen reduction calculation details.

The 2D heatmap representation of these three cases is mapped and shown in Figure 5C, where each heatmap corresponds to the data pointed by arrows shown in Figure 5B. The heatmaps reveal that the level of oxygen reduction was homogenous across the MEA surface, except where the COC and UEs are located, and at the MEA edges. The electrically inactive 80 UEs in the MEA central area created spots where less oxygen reduction occurs. Thus, it reflected as higher pO_2_ readings approximately cover 4 × 4 electrodes (110 µm × 110 µm) area around each UE. A similar pattern also existed along the edge of the MEA. Oxygen-rich media continuously diffused in from the area outside the MEA, where no oxygen reduction occurred. To increase accuracy in OCR analysis, a correction step was performed to remove the UE artifacts, and additional steps of heatmap data processing and smoothing are described in Appendix A.

The sequences of the pO_2_ heatmaps in Figure 5C indicate the capability of the electrochemical sensor system in mapping the 2D oxygen gradient, the degree of oxygen reduction by the MEA, and the respiration dynamics of COCs over time. In summary, first, the potentiostat should not be in the active state continuously for an extended period due to the limited media bulk (3.9 µL) above the MEA as indicated by case #1. One can extend the potentiostat duty cycle by increasing the microfluidic PDMS channel thickness, thus increasing the chamber volume. However, the appropriate potentiostat duty cycle is still needed to properly capture cell respiration in a relatively small volume. Second, when an appropriate potentiostat duty cycle was used (e.g., case #2), the COC respiration over time can be captured by the electrochemical sensor system as the oxygen reduction around the COC progressively expands outward over time as shown by dotted black circles in case #2. Finally, the potentiostat duty cycle should not be too small such that it loses the temporal resolution to capture respiration dynamics as shown by case #3. Therefore, to reliably measure the OCR of target cells over a desired period, the potentiostat duty cycle needs to be adjusted on a case-by-case basis to fully explore the temporal characteristics of target cells.

### 3.3. Comparison of Healthy and Dead COC

The microfluidic support system allows access to restore the pO_2_ to a saturated condition after each imaging experiment. This section discusses the results of experimental attempts to maintain oxygen-rich media flow during imaging. The experiments used healthy and dead COCs as control. The dead cells were previously exposed to media with 1% Triton™ X-100 (Sigma-Aldrich^®^, St Louis, MO, USA) for 15 min, rinsed, and placed back to the oxygen-rich media. For both cases depicted in Figure 6, the experiments were started with an injection of cells through the microfluidic inlet. Once the cells were positioned on the MEA surface, as visualized in Figure 6(Ai,Bi), a continuous stream of 40 µL/min of fresh media was held constant using a syringe pump. Any fluidic stream rate above 40 µL/min would move the cells out of the MEA area. Results presented in Figure 6 focus on the pO_2_ readings during the transition from 40 µL/min to 0 µL/min microfluidic flow, indicated as t = 0 s.

To better analyze the COC oxygen consumption signals, cross-sections in X and Y-axes of a select COC are plotted over time (Figure 6(Aii,Bii)), with the transition of blue to red curves signifying progression over time. Using the optical image as a location reference, pO_2_ activities can be observed at and around the cells. Further analysis was done by focusing on electrodes located beneath the cells. A magenta box outline was drawn around a COC in question, and their average readings were used to show signal measurement values over time, while the grey box contained a selected group of electrodes that represent background reduction current or reference measurement value. The plots of signal, reference, and their difference over time are shown in Figure 6(Aiii,Biii) with each vertical line, color-coded from blue to red, corresponding to each curve on the X and Y-axes cross-section plot. Finally, a 2D heatmap representation of each corresponding timestamp is shown accordingly in Figure 6(Aiv,Biv).

The results in Figure 6 can be thus summarized: first, in the healthy COC pO_2_ imaging, the measurement data from the cross-section in both X and Y-axes (Figure 6(Aii)) show that COC had decreasing pO_2_ towards its center. This is consistent with an early finding that the surrounding bovine cumulus cells are known to actively utilize glucose and exhibit little oxidative metabolism, whereas the oocytes at the center actively utilize OXPHOS and require only limited glucose metabolism [65]. Second, the distinct responses of pO_2_ between healthy and dead COCs were captured by the system as depicted in Figure 6(Aiii,Biii), respectively. In the healthy COC case, the pO_2_ signal reading was initially high, decreasing at a similar rate as that of the background oxygen reduction, while maintaining a constant difference of ~20 µM pO_2_ for t > 15 s. The ~20 µM difference between the signal and the reference suggests metabolic respiration by the healthy COC. Meanwhile, the dead COC case shows a trend where the signal approaches the reference within 52 s, indicating the absence of active oxygen consumption. Third, as depicted in Figure 6(Aiv,Biv), the 2D heatmaps show gradients of oxygen-rich media that were initially injected from the left side of the MEA and stopped at t = 0 s, after a previous oxygen-reduced state around t = −20 s. The heatmaps illustrate that the signal near the healthy COC shows a visible oxygen consumption in contrast with the surrounding area of the dead COC over the entire period of the experiment. Another observation that can be made from Figure 6(Biv) is that the dead COCs physically disturbed the dispersion of the oxygen-rich media and reflected as reduction signals resembling the exact location of the COCs. This disturbance is mainly due to the physical obstruction of dead COCs. The dead COCs inhibit the media flow and diffusion of oxygen-rich media to replenish the pO_2_ surrounding the dead COCs, resulting in lower pO_2_ readings in the shadow of the dead COCs due to continued oxygen reduction by the MEA. As it approaches t = 52 s, the reduction signals around COCs appear to be at the same level as the rest of the MEA area, suggesting no oxygen consumption by the COCs that result in homogenous pO_2_ across the MEA area. Therefore, to obtain accurate measurement data, the pO_2_ imaging and analysis should be performed when the microfluidic flow is at static (zero fluidic flow) or when the effects of disturbed fluidic flow and limited diffusion are already minimized (i.e., t > 26 s in our case). Appendix A shows the captured movie of heatmap presented in Figure 6A.

### 3.4. Oxygen Flux and Consumption Rate Analysis

OCR is one of the essential parameters in determining the viability of oocytes. A set of healthy COCs in static media were used to calculate oxygen flux and consumption rate. Figure 7A illustrates an expected trend of oxygen consumption of healthy COCs, comparable to pO_2_ responses in Figure 5 and Figure 6A.

The flux and rate of oxygen consumption are based on the selected Y-axis cross-section shown in Figure 7(Aii). We use the spherical diffusion theory that was previously developed and described for analyzing oocyte/embryo cells [43]. Flux density (mol/s/m^2^) of pO_2_ can be described as
(1)$$f_{s}=D\Delta C/x$$
where *D* (m^2^/s) is the diffusion coefficient of oxygen in water, approximated at 2.1 × 10^−9^ m^2^/s [43]; *x* (m) is the distance or radius of a spherical sample, and ∆*C* (mol/m^3^) is the concentration difference between the concentration at *x* and the bulk concentration (saturated pO_2_), which can be derived from the electrical current reading from each pixel and the calibration curve. The bulk concentration is calculated from the bottom 5% quantile electrical current values from the reference box at t = 0 s, representing the highest pO_2_ during an experiment. Hence, following the spherical diffusion theory, the OCR (mol/s) of a half-spherical sample can be described as
(2)$$r_{s}=S\cdot f_{s}=\left(2\pi x^{2}\right)\cdot D\Delta C/x=2\pi xD\Delta C$$
where *S* is the surface area of a half-spherical sample at a distance *x*.

In this specific experiment result, the center of a COC was selected from the Y-axis cross-section at Y = 72, which signifies the 72nd electrodes from the top-left origin coordinate, where the cell’s surface boundary, marked by magenta-colored lines in Figure 7(Ai,Aii), was selected at Y = 60 and Y = 84 for either side. Equations (1) and (2) were used to generate the flux and rate plots up to 18 electrodes (500 µm) away from the cell’s surface or 30 electrodes (825 µm) away from the cell’s center at Y = 72. Figure 7B(i,ii) show oxygen flux and consumption rate over 500 µm, to either side away from the cell’s surface. The curves in Figure 7(Bi,Bii) are color-coded to show different timestamps, the blue curve for t = 0 s and red for t = 46 s, with each curve corresponding to a single frame (90 frames in total).

The flux density is approximately 20 nmol/s/m^2^ at the cell’s surface and exponentially decays as it gets farther away from the cell; however, the flux curves increase as a function of time. This trend indicates that the pO_2_ decreased over time within the cell’s surface boundary and the lack of oxygen caused the surrounding oxygen to diffuse in, affecting the surrounding pO_2_. To better illustrate the flux change over time, six intercepts at different Y-axis coordinates, cell surface (red), 100 µm from the cell surface (green), and 500 µm from the cell’s surface (blue) to the left and right sides of the cell are plotted and shown in Figure 7(Biii,Biv), respectively. Each dot represents the intersect value of the flux curve at the specific Y coordinate. The responses are normalized to 0 at t = 0 s and linearly fitted to show good linear correlations (R = 87 and above) for all six cases. The plot illustrates the expected trend of increasing flux over time, with the strongest flux growth occurs at the surface of the COC. Finally, the oxygen consumption rate curves at the cell’s surface show ~12 fmol/s at t = 0 s and increases to ~18 fmol/s at t = 46 s. The high readings towards Y ≤ 60 were possibly due to the oxygen consumption by two COCs located around Y = 20. The measured OCR results are comparable to the results from our previous work and others using various stages of bovine oocytes and embryos, where the OCRs were measured between 5 to 30 fmol/s [11,12,32,66,67,68]. Furthermore, the OCR results for COCs measured by others from a variety of sources (mouse, bovine, and human) [9,69,70] also show a comparable range (3 to 60 fmol/s) as our measured OCR results. Appendix A shows the captured movie of heatmap presented in Figure 7.

## 4. Conclusions

This paper presents the earliest chemical imaging results of a novel electrochemical sensor system with a dense on-chip sensing array supported by electronic control circuits and data acquisition, microfluidic support system, and temperature control system for in vitro experiments. The highly integrated electrochemical sensor system allows the initial exploration of the electrochemical sensing method for probe-less and real-time chemical imaging at high spatial and temporal resolutions. 

Among many potential applications, the electrochemical sensor system and the stereomicroscope setup were used to demonstrate the system’s capabilities in imaging oxygen respiration of COCs. A custom microfluidic setup was optimized for handling COCs. Multiple in vitro experiments using bovine COC validate the capability of the electrochemical sensor system, for the first time, as a new tool to study real-time OCR of live cells in 2D and its implications to cell metabolism. As an advanced tool for quantitatively analyzing oxygen consumption, flux density and rate of the oxygen consumption of COCs in 27.5 µm spatial resolution and 4 Hz temporal resolution have been demonstrated. The high spatial and temporal resolution images of oxygen consumption provide information of the cell’s basal respiration that has not been previously demonstrated by others with CMOS-based MEA sensors or any other existing methods.

The impact of oxygen reduction by the electrodes within a small microfluidic chamber was presented, and possible remedies to minimize the effects were discussed. Furthermore, the system would benefit from electrode surface modifications to improve selectivity to oxygen [11,12,36] and to sense other metabolizing agents (glucose and lactose) [12,32]. Electrode surface modifications would also enable simultaneous neurotransmitter detections [28,71] with high spatial and temporal resolutions. Further investigations in the Pt electrode drift characteristics are necessary to support accurate analytical and prolonged biology experiments. In combinations with advanced microfluidic devices [50,51], the improvements would enable the electrochemical sensor system to monitor a broader metabolic panel with high sensitivity and selectivity at high spatial and temporal resolutions. Ultimately, these studies and further improvements could significantly benefit rapid cell screening under a variety of therapeutics.

## Figures and Tables

**Figure 1 biosensors-11-00256-f001:**
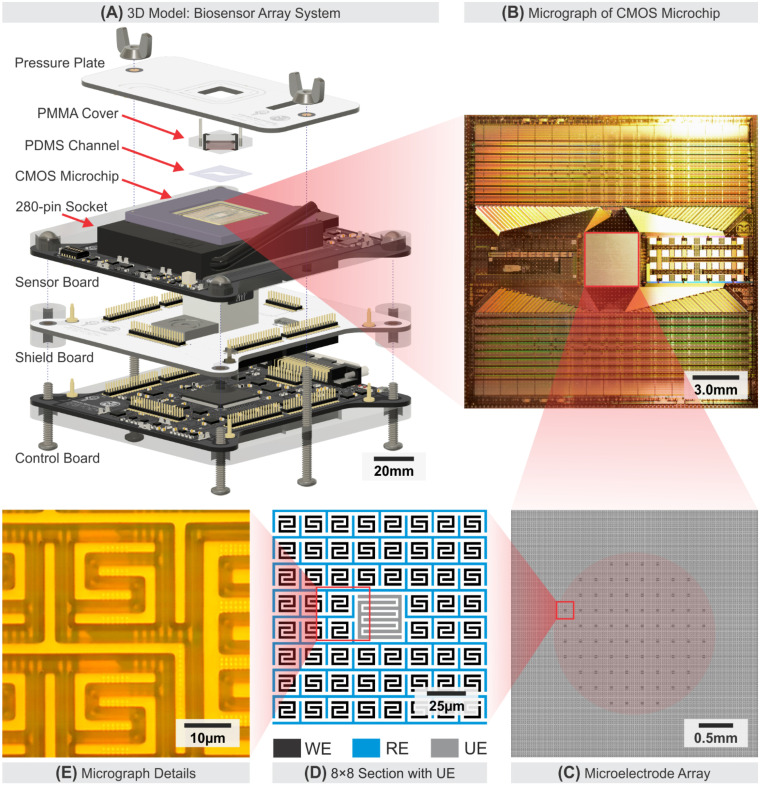
Electrochemical sensor system overview. (**A**) Sensor system 3D model assembly of the CMOS microchip, support PCBs, and microfluidic system arrangement. (**B**) CMOS microchip with MEA micrograph, showing an array of read channel circuits and the 3.6 mm × 3.6 mm MEA. (**C**) The MEA consists of 16,064 WE, an interleaved RE, and 80 pairs of interdigitated UEs around the center of the MEA. (**D**) An 8 × 8 subsection of the MEA showing configurations of the WE, RE, and UE. (**E**) A detailed micrograph of the surface electrode.

**Figure 2 biosensors-11-00256-f002:**
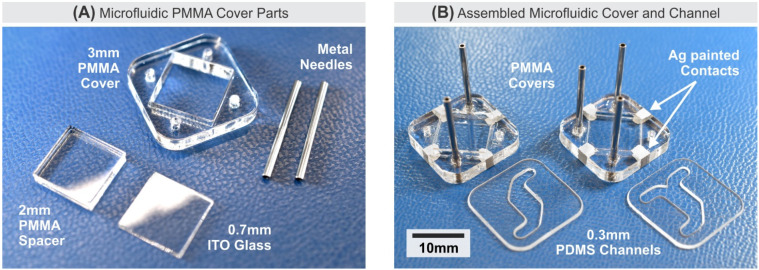
Microfluidic support system. (**A**) Unassembled parts of the microfluidic cover obtained and fabricated from low-cost off-the-shelf components and a laser cutter. (**B**) Two types of assembled microfluidic PMMA cover and PDMS channel.

**Figure 3 biosensors-11-00256-f003:**
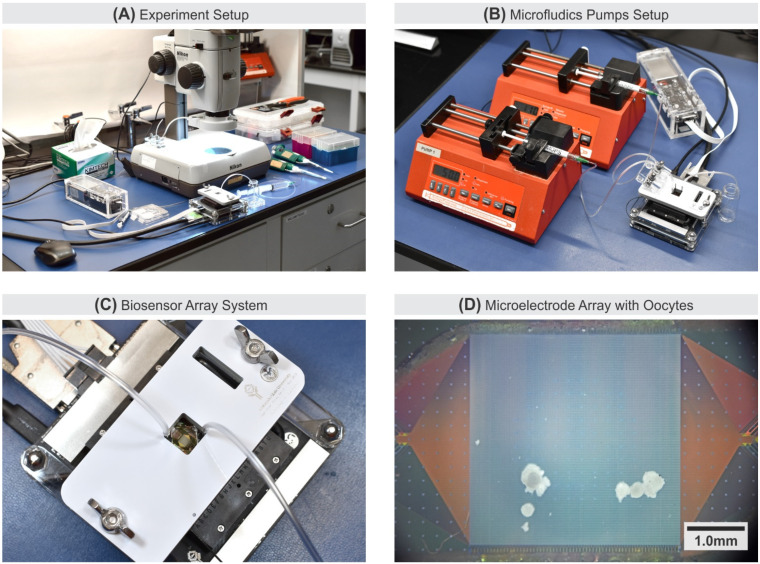
Electrochemical sensor with microfluidic system. (**A**) Typical In vitro experiment setup employing a stereomicroscope. (**B**) A setup employing two syringe pumps for simultaneous delivery of two types of solutions. (**C**) A close-up system setup showing the clear microfluidic PMMA cover with integrated ITO-coated glass, and an opening in the pressure plate allowing optical imaging of the MEA area. (**D**) An image from the stereomicroscope depicting loaded COCs to the MEA surface.

**Figure 4 biosensors-11-00256-f004:**
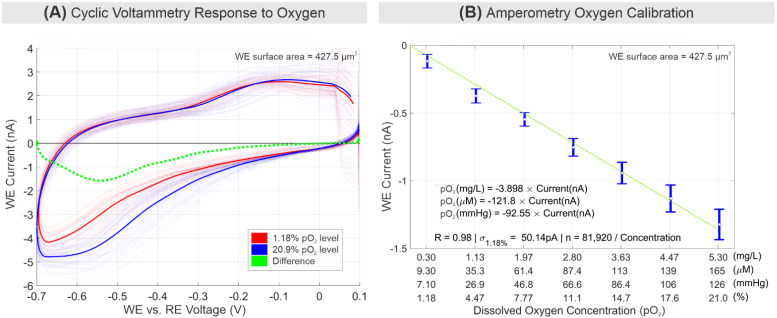
Electrochemical oxygen response. (**A**) Cyclic voltammetry comparison between media with 1.18% to 20.95% pO_2_ at 10 V/s scan rate over 64 selected WEs. Resulting readings from every 64 channels are plotted as overlapping opaque curves, and their average is plotted as a solid curve. (**B**) Amperometry sensitivity response of oxygen in media for 1.18% to 20.95% pO_2_. The error bar signifies 1-σ values around the mean.

**Figure 5 biosensors-11-00256-f005:**
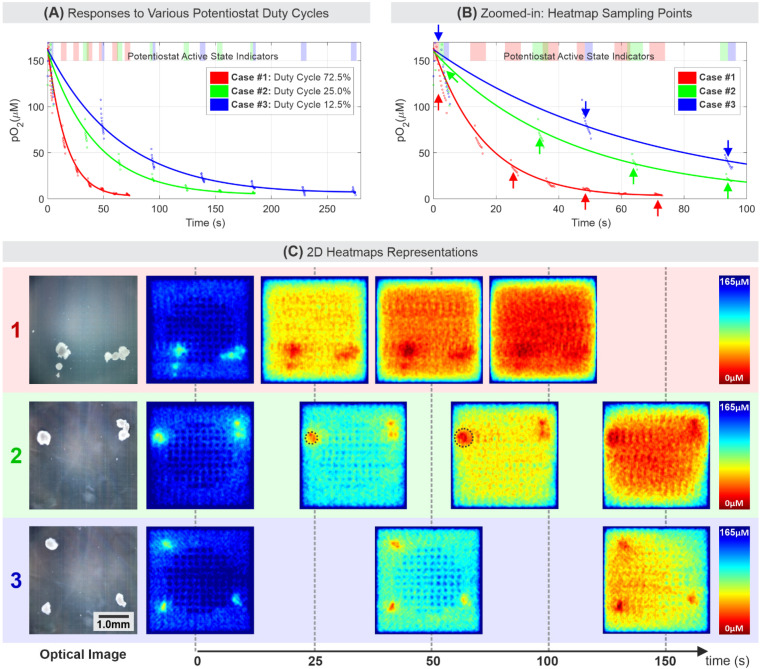
(**A**,**B**) The oxygen reduction rate of the sensor at three different potentiostat duty cycles. The potentiostat on the indicator bar signifies that the potentiostat is active (V_RE_ = −0.6 V) during the period shown by the shaded bars. (**C**) The corresponding 2D heatmap of three different cases of the in vitro experiment. Each heatmap corresponds to a 2D heatmap frame selected at a time point as pointed by the arrows in (**B**).

**Figure 6 biosensors-11-00256-f006:**
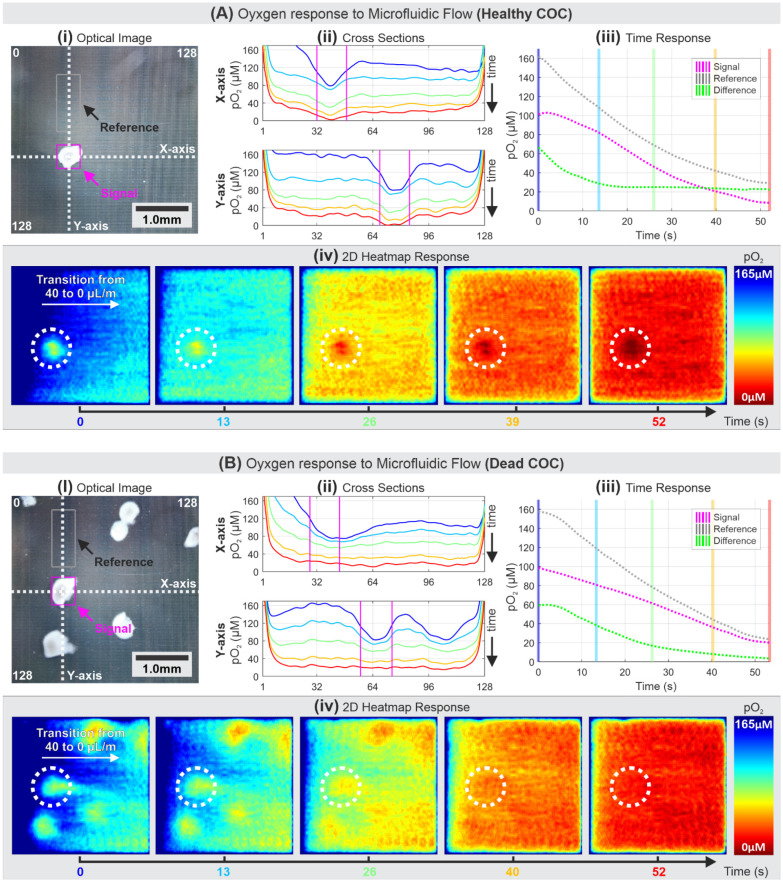
pO_2_ imaging after a transition from 40 µL/min to 0 µL/min microfluidic flow with (**A**) a healthy COC and (**B**) dead COCs. The white dotted circles compare the pO_2_ imaging response between the healthy and dead COC.

**Figure 7 biosensors-11-00256-f007:**
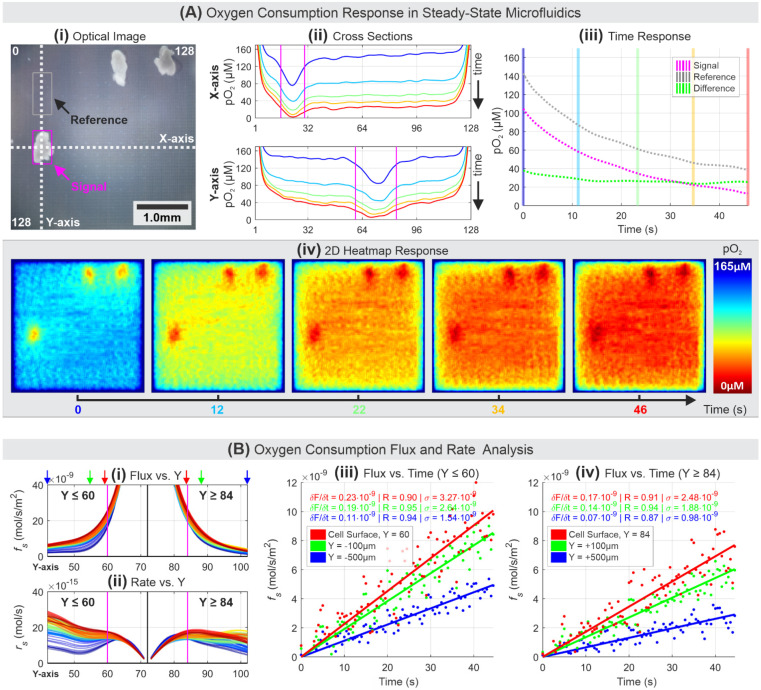
(**A**) Oxygen gradient imaging in static media of a healthy COC. (**B**) (i,ii) Analysis of oxygen consumption flux and rate based on the Y-axis cross-section at multiple distances over a period. Flux scatter plots in (iii,iv) correspond to Y locations at the surface of COC, 100 µm, and 500 µm away to either side of the COC, with each Y location pointed by arrows in (i).

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
