# Peer review of "Real-Time Analysis of Oxygen Gradient in Oocyte Respiration Using a High-Density Microelectrode Array"

_biosensors, 2021, doi:10.3390/bios11080256_

Round 1
Reviewer 1 Report
In this manuscript, the authors present an electrochemical sensor platform for oxygen sensing by fabricating a CMOS-based microchip and its surface MEA. They examined the performance of the system for cellular oxygen sensing after combining an electrochemical sensor system and a microscope and acquired corresponding images. This work shows some novelty and the manuscript is well organized, but I think additional data is required for publication as a full paper and several points are requiring further modification.
Here are my suggestions:
1、 In the introduction, the paper mentioned that the electrochemical method has advantages over others such as the optical method, but a comparison with work using the electrochemical method for oxygen detection is lacking. The authors are suggested to make a comparison with them.
2、 The electrodes may lead to biological contamination and corrosion during the experiment, while substances present in the cell culture solution may also have an impact on the results. How did the authors avoid the effect of electrical signals on the cell state in their experiments?
3、The speed of detection is important when faced with large samples in an application. How fast can this system detect the samples? Can there be accurate data to describe the speed?
Author Response
Dear Reviewer,
We greatly appreciate your time for reviewing and providing us with comments and suggestions.
Please find the revised manuscript attached as a Word file and the responses to each point below. We have also corrected language-related issues in the manuscript. We refer to the manuscript content by the line number; please disable the Microsoft Word markup to show the manuscript’s final form with the stated line number.
Here are my suggestions:
- In the introduction, the paper mentioned that the electrochemical method has advantages over others such as the optical method, but a comparison with work using the electrochemical method for oxygen detection is lacking. The authors are suggested to make a comparison with them.
Response:
The Introduction – second paragraph (starting at line 54) provides a historical overview and comparison to other methods of pO2 sensing. In the third paragraph (starting at line 75), the overview lists other electrochemical pO2 sensing works in its early development of one-dimensional electrode sensing, progressing up to the latest integration of two-dimensional electrodes array. Both paragraphs also include the other works’ achievements and drawbacks. Due to the novel nature of this concept, there are limited numbers of studies in the integration of electrochemical detection of pO2 for high two-dimensional spatial resolution while keeping a respectable fast temporal resolution.
In addition, in-depth technical comparison of the system (e.g. spatial & temporal resolutions and quantitative electrochemical sensing performances) to other similar works has been presented in our published paper. For the comparison summary, please refer to Table III in [48] Tedjo, W.; Chen, T. An Integrated Biosensor System with a High-Density Microelectrode Array for Real-Time Electrochemical Imaging. IEEE Trans. Biomed. Circuits Syst. 2020, 14, 20–35, doi:10.1109/TBCAS.2019.2953579.
- The electrodes may lead to biological contamination and corrosion during the experiment, while substances present in the cell culture solution may also have an impact on the results. How did the authors avoid the effect of electrical signals on the cell state in their experiments?
Response:
As mentioned in Section 2.2 (line 159), the MEA is coated with Pt on its outermost layer. Many other studies and long-standing applications in analytical chemistry proved that Pt coating, as an inert noble metal, provides one of the highest resistivity to chemical fouling. Additionally, experiments reported in this paper were designed to be completed within a few minutes, where the effect of degradation of Pt electrode surface is negligible. We agree that longer (days) experiments involving cell culturing may lead to electrode degradation, which could cause inaccurate readings. Techniques, such as in-situ periodic calibration of electrodes, can be employed to alleviate the impact of electrode fouling over time, however, they are beyond the scope of this paper. We have previously captured this matter in a statement for our future work in the Conclusion (line 502).
In regard to electrochemical impact from other substances present in the cell culture – we have considered and discussed this matter in detail in Section 3.1 – second paragraph (line 263). Furthermore, Section 3.2 – second paragraph (line 325) and Supporting Information 3 also discuss and relate the oxygen reduction and the amount of charge that occurs on each WE. Ultimately, to further increase selectivity to only oxygen, electrode surface modifications and a better suitable RE material (Ag/AgCl) are required. We have previously captured this discussion within Section 3.1 and in the Conclusion (line 499).
Regarding the electrical signal's effect on the cell state – the electrochemical electron transfer in a microelectrode array configuration occurs at the surface of the MEA, where the diffusion layer forms. The maximum diffusion layer thickness is expected to be in the order of micrometers or less. Therefore, the effect of the electrical signal on the bulk solution outside the diffusion layer, where the cell resides, is insignificant. This statement can be proven by the heatmap images shown in Figure 6B(iv) at t = 52s; the readings would have been affected by the existence of the dead COCs if there was any electrical current injected into the body of the cells.
- The speed of detection is important when faced with large samples in an application. How fast can this system detect the samples? Can there be accurate data to describe the speed?
Response:
In this paper, we would like to focus specifically on validating the two-dimensional electrochemical fundamentals of pO2 sensing using the integrated CMOS MEA system, and presenting the results and capabilities of the system to capture and analyze real-time pO2 readings of living COCs; both aspects have not been reported previously. We believe the methodologies presented in this paper are essential for providing a solid pathway for many other possible applications. The presented system, which offers 4 Hz temporal resolution and 27.5µm spatial resolution with a 3.6 mm × 3.6 mm detection area would certainly be a great platform for rapid detection of a large number of samples. We have previously captured this matter in a closing statement for our future work in the Conclusion (line 507).

Reviewer 2 Report
In this manuscript, Tedjo and colleagues build on their previous efforts in electrochemical imaging of oxygen using microelectrode arrays. Compared to previous publications from the same authors and the state of the art, the key novelty of this work is the relatively high spatial and temporal resolution achieved (27.5 μm and 0.25s, respectively). These features enabled the authors to map 2D gradients of oxygen generated by oocyte respiration onto the microelectrode array, which is the key result of this work. It’s also noteworthy that the sensor is highly integrated with a microfluidic delivery method and temperature control; and that the effect of flow rate, which might be detrimental to the quantification of the oxygen gradient, is also evaluated ad accounted for.
Overall, I enjoyed reading the paper, which is well written and organised logically. The methodology is appropriate and explained in sufficient detail. The claims and conclusions are also well supported by the evidence presented. For these reasons, I recommend the paper be published in biosensors after the authors address a few minor points. Please find a list below:
- It is not clear why the authors are testing the effects of the reduction of the flow rate over time. As mentioned above, I believe that taking this effect into account could be a strength of the manuscript, considering that such sensors are often exposed to moving fluids and microfluidic delivery methods; however, I wonder whether it would be possible to wait for the fluid to stabilise and then start measurement in steady state. In fact, isn’t this the case for the flux analysis conducted in figure 7? In other words, wouldn’t it be better to make the comparison between healthy and dead COC in steady state to show the differences in respiration more convincingly?
- In the current version of the heatmaps (figures 6), it is not clear what the white dotted circles around the cells represent. I assume they pinpoint the cells’ locations but, besides this, they are of little use. As a suggestion, the authors might want to consider adding iso-concentration contours around the cells, namely plot contour lines at constant pO2 concentration that might show nicely the oxygen diffusion over time.
- It does not seem clear whether any experimental repeats were performed. I think that a powerful aspect of the proposed method is the ability to map gradients from multiple single cells, therefore studying heterogeneity and cell-to-cell differences. I suggest that the authors include experimental repeats if they have performed any, to show heterogeneity or, similarly, include results from the multiple cells within the field of view, such as the extra cells in figure 7, for instance.
- The authors say that the system was calibrated with respect to room temperature. How was this done? With an external thermocouple? And how did the authors make sure that the temperature was 38 °C at the array surface?
- In the conclusions, the authors say that the device was optimised for cells above 50 μm in diameter? Why is this the case? What would be the challenges of using smaller cells? I would expect that not much would need to change if different cells were to be used.
Author Response
Dear Reviewer,
We greatly appreciate your time for reviewing and providing us with comments and suggestions. Please find the revised manuscript attached as a Word file and responses to each point below. We refer to the manuscript content by the line number; please disable the Microsoft Word markup to show the manuscript’s final form with the stated line number.
- It is not clear why the authors are testing the effects of the reduction of the flow rate over time. As mentioned above, I believe that taking this effect into account could be a strength of the manuscript, considering that such sensors are often exposed to moving fluids and microfluidic delivery methods; however, I wonder whether it would be possible to wait for the fluid to stabilise and then start measurement in steady state. In fact, isn’t this the case for the flux analysis conducted in figure 7? In other words, wouldn’t it be better to make the comparison between healthy and dead COC in steady state to show the differences in respiration more convincingly?
Response:
We greatly appreciate the reviewer’s complete understanding of our paper. At first, we conducted our experiments with slow-moving media to ensure continuous deliverable of the nutrients needed by the cells. However, we found that disturbances in the fluid affect the electrochemical readings. Therefore, briefly stopping media flow to achieve a steady-state condition prior to electrochemical reading is required to accurately perform further analysis (e.g. pO2 flux calculation presented in Figure 7).
Concerning the reading of healthy versus dead COCs in steady-state – we learned that one of the effective ways to identify dead COCs exist physically, but not consuming oxygen, was to initially have fluid movements through the dead COCs while capturing the pO2 images. Therefore, results in Figure 6B were purposely compiled to show flow disturbances due to dead COCs (t = 0 s), but dead COCs did not impact the electrochemical readings at a steady-state (t = 52 s). In other words, electrochemical reading after the fluid reaches a steady-state would only result in a homogenous reduction heatmap across the entire MEA. Referring to Figure 6B(iv), without any initial fluid movements, the heatmap colormap would start with all dark blue (t = 0 s) and end with all dark red (t = 52s). It would be challenging to convince readers if we would have to show plain heatmaps (transitioning to all blue to all read) and claim there were dead COCs.
- In the current version of the heatmaps (figures 6), it is not clear what the white dotted circles around the cells represent. I assume they pinpoint the cells’ locations but, besides this, they are of little use. As a suggestion, the authors might want to consider adding iso-concentration contours around the cells, namely plot contour lines at constant pO2 concentration that might show nicely the oxygen diffusion over time.
Response:
The reviewer’s assumptions are correct. The white markings were not included initially. However, after rounds of reviews, many readers believe that we should place markings to clearly show the difference between the two cases. It is also stated in the Figure 6 caption that the white dotted circles’ purpose is to compare between the healthy and the dead COCs. We hope to keep the marking in the figures.
In response to iso-concentration contours – we have attempted combinations of visualization types with many colormaps, with and without the iso-concentration contouring. After all, this exact colormap configuration (gradient of deep blue to deep red) gives us the best two-dimensional qualitative representation. Meanwhile, the X-axis and Y-axis cross-sections (Figure 6A(ii), 6B(ii), and 7A(ii)) provide the best two-dimensional quantitative representation.
- It does not seem clear whether any experimental repeats were performed. I think that a powerful aspect of the proposed method is the ability to map gradients from multiple single cells, therefore studying heterogeneity and cell-to-cell differences. I suggest that the authors include experimental repeats if they have performed any, to show heterogeneity or, similarly, include results from the multiple cells within the field of view, such as the extra cells in figure 7, for instance.
Response:
We agree with the reviewer’s statement. In this paper, we would like to focus specifically on validating the two-dimensional electrochemical fundamentals of pO2 sensing using the integrated CMOS MEA system, and presenting the results and capabilities of the system to capture and analyze real-time pO2 readings of living COCs; both aspects have not been reported previously. We believe the methodologies presented in this paper are essential for providing a solid pathway for many other possible applications. The presented system, which offers 4 Hz temporal resolution and 27.5µm spatial resolution with a 3.6 mm × 3.6 mm detection area would certainly be a great platform for rapid detection of a large number of samples. We have previously captured this matter in a closing statement for our future work in the Conclusion (line 507).
- The authors say that the system was calibrated with respect to room temperature. How was this done? With an external thermocouple? And how did the authors make sure that the temperature was 38 °C at the array surface?
Response:
For initial calibration, we attached a highly accurate and calibrated thermometer (±0.05 °C accuracy, 0.01 °C resolution). The thermometer probe tip, where the measurement occurs, was coated with a thermally conductive pad and held down to the electrode array surface of the microchip to obtain a localized steady-state reading at the array surface area. Our laboratory temperature has been well controlled between when the initial calibration and experiments were conducted. For an extra layer of assurance, a thermal imaging camera was periodically used to confirm whether the array surface is within the tolerable range. The product information of the external thermometer and thermal imaging devices used for calibration were added (lines 222 and 226).
- In the conclusions, the authors say that the device was optimised for cells above 50 μm in diameter? Why is this the case? What would be the challenges of using smaller cells? I would expect that not much would need to change if different cells were to be used.
Response:
Due to the 27.5 μm spatial resolution, it would be a challenge to accurately analyze and differentiate smaller size cells. Also, due to the required small focal length, the existing microfluidic cover design does not allow high magnification optics to be used, therefore, the 50 μm cell diameter claim was made. We agree with the reviewer’s comment, regardless of the two limitations listed above, the users may have different perspectives and expectations over various applications (e.g. imaging clusters of smaller size cells). To avoid confusion, all statements related to 50 μm cell diameter (lines 138 and 488) were removed from the manuscript.
Reviewer 3 Report
Interesting paper, paper presents an application of an electrochemical sensor platform with a custom‐designed CMOS‐based microchip and its Pt‐coated surface MEA.
Presented system in the future with potential improvements could significantly benefit rapid cell screening under a variety of therapeutics.
Cosmetic - “enter” line 46.
All section are adequate, all: introduction, system, methodology, setup, measurements, results, literature are clearly presented with high scientific soundness.
I suggest to accept in present form.
Author Response
Dear Reviewer,
We greatly appreciate your time for reviewing and providing us with your positive response. The cosmetic error at line 46 has been corrected. Please find the final revised version of the manuscript attached as a Word file.
Round 2
Reviewer 1 Report
The authors have answered my questions in detail and this paper can be published